# A systematic review and meta-analysis on pharmacist-led interventions for the management of peptic ulcer disease

Biswash Sapkota[1]*, Bipindra Pandey[1], Bishal Sapkota[2], Keshav Dhakal[3], Bijay Aryal[4]

1 Department of Pharmacy and Clinical Pharmacology, Madan Bhandari Academy of Health Sciences, Hetauda, Nepal, 2 Department of General Practice and Emergency Medicine, Patan Academy of Health Sciences, Lalitpur, Nepal, 3 Department of Pharmacy, Little Buddha Academy of Health Sciences, Kathmandu, Nepal, 4 School of Pharmacy, Karnali Academy of Health Sciences, Karnali, Nepal

* biswash.sapkota@mbahs.edu.np

## Abstract

Pharmacists are essential for developing pharmacotherapy plans, conducting clinical assessments, and overseeing drug monitoring. Their interventions help prevent medication errors and adverse drug events and enhance medication safety. This study aimed to systematically review pharmacist-led interventions for managing medication-related issues in patients receiving anti-ulcer treatments. A systematic review and meta-analysis was performed to explore four databases for studies published from 1904 up to June 2024. Nine studies were reviewed, including four retrospective, three case-control, one mixed-method, and one prospective pre-post study involving 34,099 participants. The average age of the patients was 61 years, and 50.23% were male. The study quality was high, with an average score of 6.22/7 on the modified Newcastle-Ottawa scale. All studies involved direct interactions between pharmacists and patients or physicians, and data were primarily collected from hospital electronic records. Pooled analysis demonstrated that pharmacist interventions significantly improved the rational use of anti-ulcer medications (OR: 4.5; 95% CI: 0.97 to 20.80; $I^2$ = 89%, P = 0.05), as reported by studies. Pharmacist interventions have a significant impact on improving rational drug use, reducing costs and treatment duration, and enhancing appropriate medication use. These interventions also positively influenced medication adherence and the correction of irrational drug use.

## Introduction

Peptic ulcer disease (PUD) is characterized by damage to the lining of the gastrointestinal (GI) tract and is primarily caused by gastric acid and pepsin. While it commonly affects the stomach and upper duodenum, it can also affect the lower esophagus, distal duodenum, and jejunum [1]. Patients with stomach ulcers typically experience epigastric discomfort 15-30 minutes after eating, whereas those with duodenal ulcers feel pain 2-3 hours post-meal. All patients with PUD should undergo *Helicobacter pylori* testing, and endoscopy may be necessary for a definitive diagnosis, especially in cases with symptoms. Most patients respond favorably to a triple therapy regimen involving proton pump inhibitors (PPIs) [2,3]. PUD is

**Data availability statement:** All relevant data are within the manuscript and its Supporting Information files.

**Funding:** The author(s) received no specific funding for this work.

**Competing interests:** The authors have declared that no competing interest exist.

a worldwide health concern, with a 5%-10% lifetime risk [4,5]. Successful treatment requires appropriate medication use and oversight by healthcare professionals, including doctors, nurses, and pharmacists.

According to Gary H. Smith's 1971 pharmacist study has significant importance as a healthcare practitioner in the management of patients with peptic ulcers. Pharmacists can provide individual patient instructions on prescribed prescription usage and collaborate with physicians on drug therapy, including drug selection, potential side effects, and drug interactions [6]. Thus, pharmaceutical intervention care is crucial for disease-treatment procedures. Pharmacists play a recognized role in the selection of pharmacotherapies and in making judgments regarding the assessment of clinical parameters and monitoring of medication [7,8]. Pharmacist interventions are crucial in preventing medication errors (MEs) and adverse drug events (ADEs), thereby improving medication safety. Additionally, their involvement in intensive care units (ICUs) and collaboration with ICU physicians have been shown to lower drug consumption, leading to reduced drug therapy costs, shorter hospital stays, and prevention of inappropriate drug use or ADEs, thereby avoiding associated expenses [9,10].

To date, no systematic study has been published that highlighted the changing role of pharmacists in the management of peptic ulcers. Therefore, this study aimed to thoroughly review and summarize pharmacist interventions to address medication-related issues in patients receiving anti-ulcer treatments. Based on the evidence from this review, healthcare systems may benefit from integrating pharmacists into care teams for peptic ulcer management.

## Methods

### Protocol and registration

This systematic review was prospectively registered with PROSPERO (registration no. CRD 42024525479). It follows the guidelines of the Preferred Reporting Items for Systematic Reviews and Meta-Analyses (PRISMA) [11,12].

### Eligibility criteria

Studies were included if they met the following criteria: (a) observational research (e.g., cohort, case-control, cross-sectional, and case series studies) and retrospective studies with clear, extractable data on patient interventions; (b) involved pharmacist-led interventions, either individually or as part of a multidisciplinary team (including medication review, pharmaceutical care, patient education, or counseling); (c) targeted adult patients aged ≥ 18 years; and (d) included patients who were using at least one anti-ulcer medication. All study designs were included in this study.

The exclusion criteria were (a) review articles, non-research letters, editorials, commentaries, animal studies, original research with fewer than 10 samples, abstracts from meeting proceedings, and articles not published in English; (b) studies focusing exclusively on children or pregnant women; (c) studies lacking full-text versions; and (d) articles that were not peer-reviewed or not accepted for publication.

### Search strategy and selection criteria

A comprehensive systematic literature search was conducted in PubMed, Medline, Europe PMC, and Google Scholar from March 17, 2024, to June 13, 2024, to gather relevant data published from 1904 through June 2024. These databases were searched using the following search terms: ("Peptic ulcer" OR" Peptic ulcer disease" OR" Gastric ulcer" OR "Duodenal ulcer" OR" Gastritis" OR " Gastroesophageal reflux disease") AND ("Pharmaceutical care" OR " Patient counseling" OR " Medication errors" OR "Drug interactions "OR " Medication

reviews" OR " Associated factors") (S1 File). After the initial search, duplicates were deleted, and two reviewer (BS and BP) separately assessed the titles and abstracts for potentially relevant papers. The complete texts of the relevant papers were checked for eligibility requirements. The reference list of acceptable research and pertinent systematic reviews was also examined to reduce literature omissions.

## Data extraction and quality assessment

Two reviewers (BS and BP) independently collected the following information from each included study: first author's name, country of origin, publication year, study location, type of study, data collection dates, gender, age, number of cases in each group, details of the interventions, and study conclusions. A third reviewer (KD) reviewed the article list and verified the extracted data to ensure the absence of duplicate articles or repeated information.

The modified Newcastle Ottawa Scale (NOS) [8] was used to evaluate the risk of bias in the research [13,14]. NOS assesses study quality based on a total of 7 points. A study receiving all seven points was considered to be of high quality, while fewer points indicated lower quality. For each original study included, the quality assessment was performed independently by two reviewers (BS and BA), and any disagreements were resolved through discussion with additional reviewers.

## Statistical analysis

A modified version of the DEPICT tool (version 2) [15] was used to categorize pharmacist interventions. The study findings are displayed in tables and elaborated upon in the results section. To assess the three primary outcomes, a meta-analysis was conducted: the effect of pharmaceutical interventions on the rational use of anti-ulcer drugs, anti-ulcer drug use for Stress Ulcer Prophylaxis (SUP), and therapy duration. When comparable outcome data from at least two studies were available, the information was aggregated. Owing to anticipated clinical and methodological heterogeneity, a random-effects model was employed. For categorical variables, results were expressed as odds ratios, whereas standard mean differences (SMD) were used for continuous variables, both with 95% confidence intervals. The $I^2$ statistic was used to evaluate the statistical heterogeneity. Subgroup analyss were performed based on pharmacist activities, rational medication use, and duration of therapy. Review Manager Version 5.4.1 was used for all analyses.

# Results

## Search results

The study selection process is illustrated in Fig 1. This study followed the PRISMA flow diagram and included four stages: identification, screening, eligibility, and inclusion. The initial database search yielded 5,616 articles; after removing duplicates 5,431 articles remained. The two reviewers evaluated these publications by examining their titles and abstracts. Ultimately, 11 articles were included in the review, although 2 were excluded due to the unavailability of full texts. Finally, nine studies were included in the investigation.

## Analysis of included articles

Nine articles were reviewed, including four retrospective studies, three case-control studies, one mixed-method study, and one exploratory prospective pre- and post-study. These studies were conducted in various countries: the United States (n = 3), China (n = 2), Canada (n = 1), Italy (n = 1), France (n = 1), and Iran (n = 1). Eight of the studies were conducted

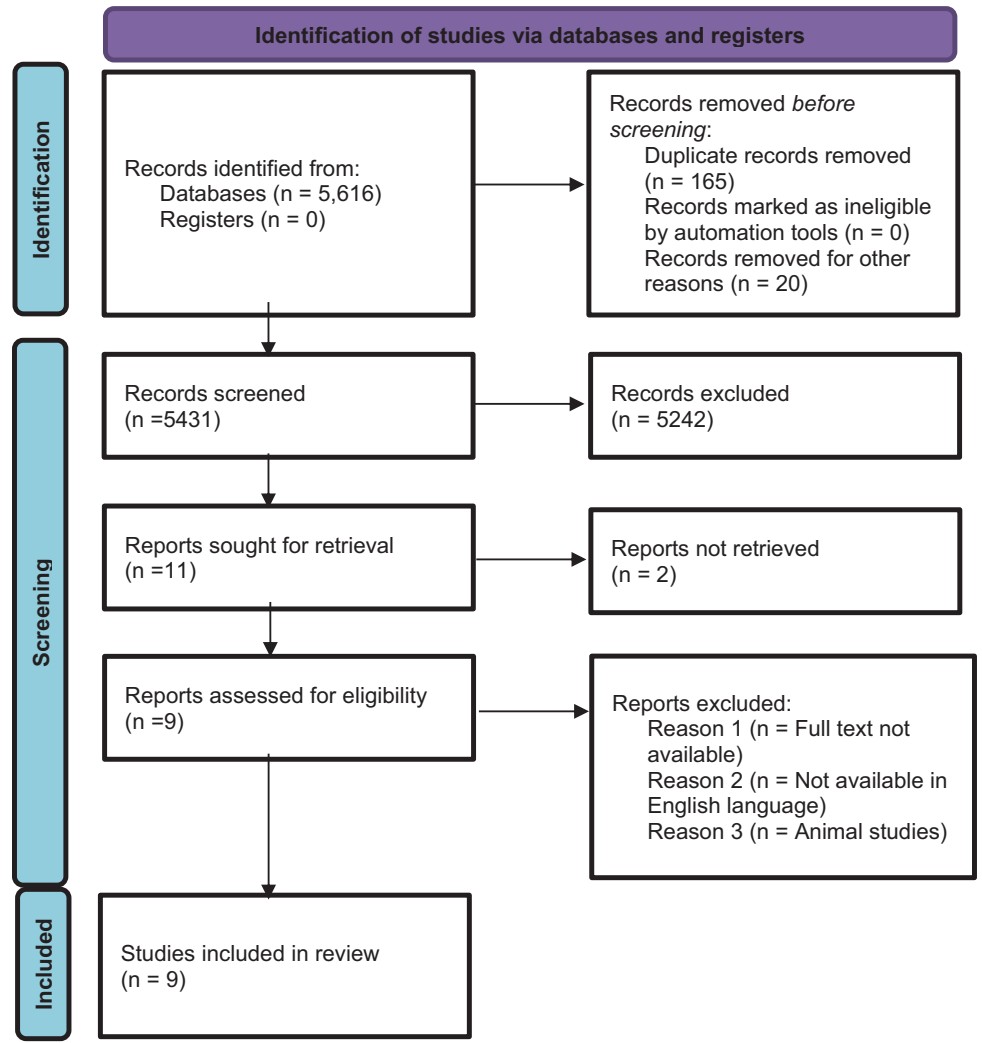

**Fig 1. PRISMA flow diagram.**

in hospitals, while one was conducted with discharged patients via telephone. The studies collectively involved 34,099 participants, with a median sample size of 502 (range, 58–29,694). The average age of the participants was 61 years and more than half (50.23%) were male. The participants used PPIs, acid suppression treatments (AST), triple or quadruple regimens, and various anti-ulcer medications. The characteristics of the reviewed articles are listed in Table 1. All studies described pharmacist-delivered services and evaluated outcomes related to pharmacist interventions.

## Quality of the included articles

The average score of the included articles was 6.22 out of seven, according to the modified NOS (Table 2). All studies were classified as medium- or high-quality studies. Four articles did not meet the criteria for sample representativeness (sample size ≥ 1000 participants). Three articles did not report non-respondents or had a response rate of less than 80%. All other criteria were met by articles. The quality assessment results of this systematic review are presented in Table 2.

**Table 1. Characteristics of included studies.**

| Name of author and year | Study design and Study site | Study duration | Gender (M/F) | Sample size | Average age (years) | Objectives | Inclusion criteria | Reference |
|---|---|---|---|---|---|---|---|---|
| Xin et al., 2018 | Case-control study, Tongde Hospital of Zhejiang province, China | February 1, 2017 - July 31, 2017 | 118/167 | 285 | 59 | To assess whether pharmacological interventions significantly impact the rational use of PPIs and reduce medication costs | Patients admitted to the hospital during the study period, who were at least 18 years old and had current orders for PPIs along with recent laboratory test results | [16] |
| Yailian AL et al., 2022 | Retrospective study, French Hospital | January 1, 2007, to December 31, 2019 | 15011/14683 | 29694 | NR | To evaluate the role of pharmacists in PPI deprescribing, analyze medication-related issues related to PPI prescriptions, examine the associated pharmacist interventions, and identify factors influencing physician acceptance of pharmacist recommendations | Medication order evaluations were conducted over a 12-year period. Within the complete database, pharmacist interventions related to PPIs were extracted. | [17] |
| Chen Q et al., 2022 | Case-control study, General Surgery Department, Fuijan Medical University Hospital, China | June and November, 2018 and between June and November, 2019 | 495/579 | 1074 | 53 | To determine if pharmacist interventions could increase the rate of rational PPI use and reduce the pharmaceutical burden on patients | Patients aged 18 and older admitted to the surgery department with diagnoses related to the esophagus, stomach, duodenum, colon, rectum, small intestine, appendix, hepatobiliary and pancreatic systems, spleen, breast, thyroid, or abdominal hernias | [18] |
| Musuuza JS et al., 2021 | mixed-methods, University of Wisconsin Health Science Hospital, USA | September 2019 to August 2020 | 150/4 | 155 | 71 | To tackle the continuous need for effective strategies that encourage guideline-adherent PPI use and minimize inappropriate use, a deimplementation intervention was created and its feasibility was assessed | Adult patients, who are not in the ICU and have an active PPI prescription, admitted to a medical or surgical unit | [19] |
| Weng A et al., 2022 | Case-control study, Telephone intervention, Putian University, Hospital, China | August 2018 to March 2019 | 80/28 | 108 | 45.5 | To explore the effects of pharmaceutical follow-up on patients with peptic ulcers who have been discharged, examining whether clinical pharmaceutical services can increase patients' understanding of peptic ulcer disease, improve adherence to treatment, and ultimately enhance treatment outcomes and quality of life | Patients aged 18 to 70 years who were discharged from the hospital during the study period with a diagnosis of peptic ulcer verified by electronic gastroscopy | [20] |
| Atkins R et al., 2013 | Retrospective study, Passavant Area Hospital, Jacksonville, Illinois | July 2010 to March 2012 | 432/657 | 1089 | 66.5 | To assess the impact of pharmacy-led training for medical staff on PPI therapy | Retrospective data were gathered from the hospital for patients who were admitted and underwent PPI therapy during the study period | [21] |
| Khalili H et al., 2010 | Exploratory prospective pre- and post-intervention, Imam Khomeini Hospital, Iran | August 1, 2008, to December 1, 2008) and post-intervention (February 1, 2009, to June 1, 2009) | 263/239 | 502 | NR | To evaluate the effect of a clinical pharmacist intervention on promoting and educating about SUP guidelines, and its impact on the appropriate use of AST | Data were gathered from all patients admitted to the infectious disease ward, with a focus on SUP and physician prescribing practices for AST | [22] |
| Buckley et al., 2015 | Retrospective study, ICU and general ward | January 1, 2011, and January 31, 2012 | 568/566 | 1134 | 57 | To evaluate the clinical and economic effects of a new pharmacist-managed stress ulcer prophylaxis program for patients in both the ICU and general ward | All adult patients (aged 18 and older) who were hospitalized and received either an H2 receptor antagonist (H2RA) or a PPI were eligible for inclusion | [23] |

*(Continued)*

**Table 1.** (Continued)

| Name of author and year | Study design and Study site | Study duration | Gender (M/F) | Sample size | Average age (years) | Objectives | Inclusion criteria | Reference |
|---|---|---|---|---|---|---|---|---|
| Tandun et al., 2019 | Retrospective study, Fraser Health Authority (FHA), British Columbia | June 6 to November 12, 2018 | 14/44 | 58 | 80 | To describe the changes and factors influencing a pharmacist-led intervention for deprescribing PPIs in long-term care facilities | A report on residents with active prescriptions for any dosage of a PPI | [24] |

NR = Not reported.

**Table 2. Quality scores assessing the risk of bias using a modified Newcastle-Ottawa scale.**

| Authors | Representativeness of the sample | Sample size | Non-respondents | Ascertainment of the exposure | Comparability of subjects in different outcome groups (control for confounding) | Assessment of the outcome | A statistical test is appropriate | Total |
|---|---|---|---|---|---|---|---|---|
| Xin et al., 2018 | 0 | 1 | 0 | 1 | 1 | 1 | 1 | 5 |
| Yailian AL et al., 2022 | 1 | 1 | 1 | 1 | 1 | 1 | 1 | 7 |
| Chen Q et al, 2022 | 1 | 1 | 1 | 1 | 1 | 1 | 1 | 7 |
| Musuuza JS et al., 2021 | 1 | 1 | 1 | 1 | 1 | 1 | 1 | 7 |
| Weng A et., 2022 | 0 | 1 | 0 | 1 | 1 | 1 | 1 | 5 |
| Atkins R et al., 2013 | 1 | 1 | 1 | 1 | 1 | 1 | 1 | 7 |
| Khalili H et al., 2010 | 0 | 1 | 0 | 1 | 1 | 1 | 1 | 5 |
| Buckley et al., 2015 | 1 | 1 | 1 | 1 | 1 | 1 | 1 | 7 |
| Tandun et al., 2019 | 0 | 1 | 1 | 1 | 1 | 1 | 1 | 6 |

(Score: 1 = achieved, 0 = not achieved).

## Characteristics of pharmacist-delivered services

Table 3 presents an overview of pharmacist-delivered services as outlined in DEPICT version 2 [15,25]. Except for Yailian et al. and Atkins et al., [21] all studies involved direct, one-on-one interactions between pharmacists and patients or physicians throughout the study period. Yailian et al., collected data retrospectively from the French Act IP database, whereas Atkins et al., collected data through telephone communication with discharged patients [17]. The pharmacists' activities included discontinuing medications, switching therapies, conducting educational sessions on PPI guidelines, and monitoring drug administration. Key tools employed in these interventions included prescription audits, rational drug use software, and PPI review guidelines[16,24].

## Outcomes of pharmacist-delivered services

The first significant outcome of pharmacist-delivered services was improvement in the rational use of anti-ulcer drugs. Xin C et al., 96.5%, [16] Chen Q et al., 68.45% [18] and Weng A et al., reported 96. 25% [14] of all interventions detected irrational use of drugs, such as not taking through proper route, proper dose, proper time, and discontinuation with consulting. Pooled estimates of these three studies found that pharmacist involvement improved rational use of anti-ulcer drugs among patients as compared to control (OR: 4.5; 95% CI: 0.97 to

**Table 3.** Description of pharmacist intervention according to DEPICT version 2.

| Author | Recipients | Mode of Contact with the Recipient | Clinical data source | Classification of interventions | Pharmacist actions | Timing of pharmacist actions | Changes in therapy and lab test reported |
|---|---|---|---|---|---|---|---|
| Xin et al., 2018 | Physicians and patients hospitalized with active PPI orders | Individual, face-to-face engagement with physicians and patients | Computerized patient record system of hospital | The clinical pharmacist conducted educational group sessions on the rational use of PPIs twice a month, in addition to the regular daily rounds | The discussion with physicians and patients covered the indications for PPIs, appropriate dosages, routes of administration, updated PPI guidelines, and potential risks of adverse effects and drug interactions | Patients received standard care supplemented with pharmaceutical interventions during daily round visits | Yes |
| Yailian AL et al., 2022 | NR | NR | French act IP database | All PPI-related pharmaceutical interventions were classified according to the Anatomical Therapeutic Chemical (ATC) system. | The pharmacist interventions included switching medications, followed by discontinuation of drugs | NR | Yes |
| Chen Q et al, 2022 | Healthcare staff and patients admitted to the surgical wards | Individual, direct interaction with medical staff and patients | The complete medical data obtained from general surgery department | The review guidelines for PPIs from the Second Affiliated Hospital of Fujian Medical University were used for evaluation | A quarterly lecture on the rational use of PPIs was delivered to medical staff. Additionally, during rounds, medication indications, drug selection, administration frequency, treatment duration, and drug combinations were discussed. The rational drug use software and prescription audits were used to analyze irrational PPI use, drug costs, and observed PPI-related adverse drug reactions (ADRs) | The pharmacist intervened in treatment based on routine daily diagnoses | Yes |
| Musuuza JS et al., 2021 | Patients admitted to the surgical and medical units | One to one contact with patients | The complete record was obtained from their medical record | PPI guideline concordant were developed for intervention | This intervention involved PPI education, with patients possibly agreeing to reduce their PPI use, switch to an alternative non-PPI therapy, or use PPIs only as needed. Following discharge, the pharmacist also conducted follow-up calls to assess tolerance and continue tapering if appropriate | NR | Yes |
| Weng A et al., 2022 | Patients who were discharged from the hospital | Telephone follow up | Medication record from hospital | Awareness of fundamental ulcer knowledge, assessment of treatment adherence, evaluation of the clinical treatment plan, and *H. pylori* testing | Patients received standard discharge services, followed by pharmacist education and regular follow-up calls | One week and three weeks after discharge, as well as at the end of the medication course | Yes |
| Atkins R et al., 2013 | Medical staff | One to one contact with medical staff | Electronic medical record | Guidelines created by the Eastern Association for the Surgery of Trauma and the American Society of Health-System Pharmacists | Pharmacy staff conducted educational training on PPI use, and the duration of PPI usage was monitored | Three medical staff meeting | Yes |
| Khalili H et al., 2010 | Patients admitted to the infectious disease ward | One to one contact | Medication record from hospital | The American Society of Health-System Pharmacists' SUP guidelines | Pharmaceutical care clinic was established | NR | Yes |
| Buckley et al., 2015 | Patients in the ICU and general ward | One to one contact | Medication record from hospital | clinical pharmacist-managed program | According to the protocol, pharmacists were authorized to discontinue any acid suppression therapy that lacked a valid indication or when major risk factors were resolved in ICU patients | NR | Yes |

*(Continued)*

**Table 3.** (Continued)

| Author | Recipients | Mode of Contact with the Recipient | Clinical data source | Classification of interventions | Pharmacist actions | Timing of pharmacist actions | Changes in therapy and lab test reported |
|--------|-----------|-----------|-----------|-----------|-----------|-----------|-----------|
| Tandun et al., 2019 | Physician and patients | One to one contact with physician | Meditech, an electronic health records system | Deprescribing strategies for PPIs involved personal consultations with each physician, and included options such as abrupt discontinuation with monitoring, gradual dose reduction, switching to as-needed ranitidine, or switching to scheduled ranitidine | Drug use evaluation reports were generated in the first and third months. This was followed by in-person discussions with physicians on-site, as well as fax communications with physicians who were mainly off-site. Additionally, there was follow-up with prescribers and coordination with a second pharmacist. | NR | Yes |

NR = Not reported.

20.80; $I^2$ = 89%, P = 0.05, Fig 2). Subgroup analysis revealed that *H. pylori* eradication yielded a slightly higher effect (0.18) than non-eradication (0.16). Interventions by pharmacists significantly improved rational medication use, with a greater effect observed in direct patient interventions (risk difference = 9.68) than in lectures (1.49). The age analysis showed higher effects in participants aged < 50 years (7.12) than in those aged > 50 years (1.91) (Table 4).

The second outcome was a reduction in treatment cost and duration of therapy. Xin C et al., [16] Chen Q et al., [18] and Buckley et al., reported [23] reduction of cost and duration of treatment due to the pharmacist intervention. Pooled estimates of these three studies found that pharmacist involvement decreased the duration of therapy among patients as compared to control (SMD: -0.72; 95% CI: -1.16, -0.29; $I^2$ = 90%, P = 0.001, Fig 3). Subgroup analysis revealed a larger effect in studies within 6 months (SMD = -0.96) and single-center studies (SMD = -0.96). Longer studies (>6 months) and multicenter studies showed moderate effects (SMD = -0.51). Therapy duration was moderately reduced in the general wards (SMD = -0.61), but data for the surgery wards were not estimated (Table 5).

The third outcome was improvement in the appropriate drug for SUP. Atkins et al., [22] Khalili et al., [21] reported the significant improvement of use of PPI/ acid supression therapy for SUP. Pooled estimates of these two studies found that pharmacist involvement improved rational use of antiulcer drugs among patients as compared to control (OR: 0.71; 95% CI: 0.41 to 1.24; $I^2$ = 0%, P = 0.23, Fig 4). These studies indicated that physicians accepted pharmacist interventions at rates between 79.9% and 89.4% [16,17]. The detailed outcomes of the included studies are presented in Table 6.

## Discussion

The primary aim of our study was to conduct a systematic review of the literature to examine the nature and extent of research evaluating the effects of pharmacist-provided care in patients taking ulcer medications. Our findings revealed that most pharmacist interventions focused on ensuring rational use for improved patient outcomes. However, the delivery of these interventions varies, including approaches such as medication reviews, implementation of pharmaceutical care plans, adherence to standard guidelines, and the development of evaluation protocols.

Our study underscores the crucial role that pharmacists play in improving the rational use of anti-ulcer drugs. Pharmacist-delivered services significantly enhance appropriate

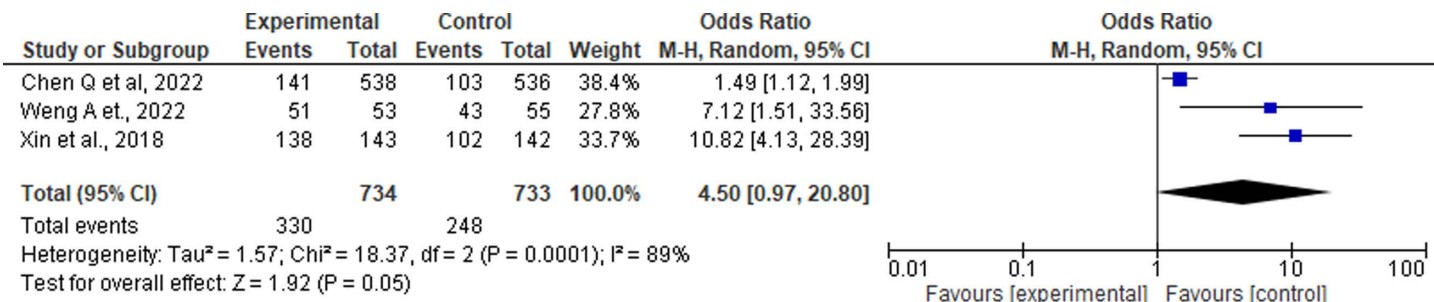

**Fig 2. The forest plot illustrates the Odds Ratio for the rational use of anti-ulcer drugs among the experimental and control participants.** The size of the data markers represents the weight from the random effects analysis and CI denotes the confidence interval.

**Table 4. Subgroup analysis on rational use of medication:** *Helicobacter pylori* eradication, intervention types, and age.

| Outcome/ source | No of studies | Number of participants | Statistical methods | Effects size (95% CI) |
|---|---|---|---|---|
| Rational use of medication | 3 [16,18,20] | 1467 | Risk difference (M-H, random, 95% CI) | 0.16 (0.04, 0.29) |
| 1.   Subgroup: By *H. pylori* eradication | | | | |
| 1.1. Eradication of *H. pylori* | 1 [20] | 108 | Risk difference (M-H, random, 95% CI) | 0.18 (0.06, 0.30) |
| 1.2. No eradication of *H. pylori* | 2 [16,18] | 1359 | Risk difference (M-H, random, 95% CI) | 0.16 (-0.02, 0.33) |
| 2.   Subgroup: By types of intervention | | | | |
| 2.1. Lecture to medical staffs and patients by pharmacist | 1 [18] | 1074 | Risk difference (M-H, random, 95% CI) | 1.49 (1.12, 1.99) |
| 2.2. Intervention to patients by pharmacist | 2 [16,20] | 393 | Risk difference (M-H, random, 95% CI) | 9.68 (4.27, 21.93) |
| 3.   Subgroup: By age of participants | | | | |
| 3.1. Age below 50 years old | 1 [20] | 108 | Risk difference (M-H, random, 95% CI) | 7.12 (1.51, 33.56) |
| 3.2. Age above 50 years old | 2 [16,18] | | Risk difference (M-H, random, 95% CI) | 1.91 (1.46, 2.50) |

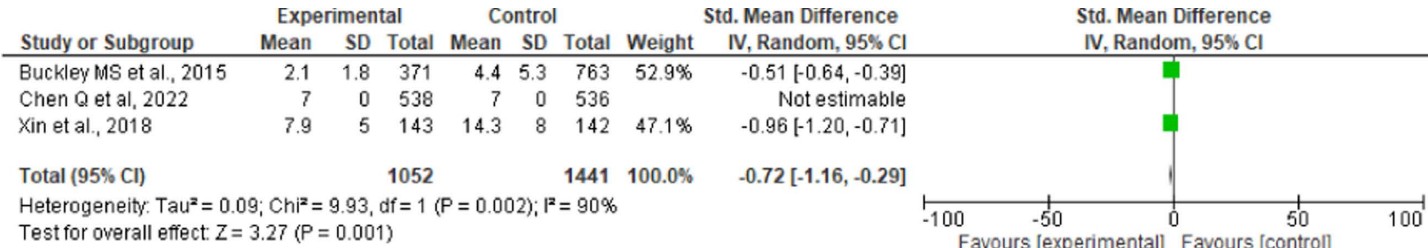

**Fig 3. The forest plot displays the Standardized Mean Difference (SMD) in the duration of therapy between the experimental and control participants.** The size of the data markers reflects the weight from the random-effects analysis, and CI denotes the confidence interval.

medication use, as reported by Xin et al. (96.5%) and Chen et al. (68.45%) and Weng et al. (96.25%) [16,18,20]. These interventions primarily address irrational drug use, including improper administration routes, incorrect dosages, and discontinuation without consultation. A pooled analysis further supported the positive impact of pharmacist involvement, with an odds ratio of 4.5, demonstrating significant improvement in the intervention group compared with the control group. This aligns with Nassir et al., (2024), in which clinical pharmacists improved the rational use of intravenous paracetamol and reduced

**Table 5. Subgroup analysis on duration of therapy: Duration of study, Center of study and wards in hospital.**

| Outcome/ source | No of studies | Number of participants | Statistical methods | Effects size (95% CI) |
|---|---|---|---|---|
| Duration of therapy | 3 [16,18,23] | 2493 | SMD (IV, random, 95% CI) | -0.72 [-1.16, -0.29] |
| 1. Subgroup: By duration of study | | | | |
| 1.1. Within 6 months | 1 [16] | 285 | SMD (IV, random, 95% CI) | -0.96 [-1.20,-0.71] |
| 1.2. More than 6 months | 2 [18,23] | 2208 | SMD (IV, random, 95% CI) | -0.51 [-0.64,-0.39] |
| 2. Subgroup: By center of study | | | | |
| 2.1. Single Center | 2 [16,23] | 1,359 | SMD (IV, random, 95% CI) | -0.96 [-1.20,-0.71] |
| 2.2. Multiple Center | 1 [18] | 1,134 | SMD (IV, random, 95% CI) | -0.51 [-0.64,-0.39] |
| 3. Subgroup: By wards in hospital | | | | |
| 3.1. General and other wards | 2 [16,23] | 1,419 | SMD (IV, random, 95% CI) | -0.61 [-0.72,-0.49] |
| 3.2. Surgery ward only | 1 [18] | 2,493 | SMD (IV, random, 95% CI) | Not estimated |

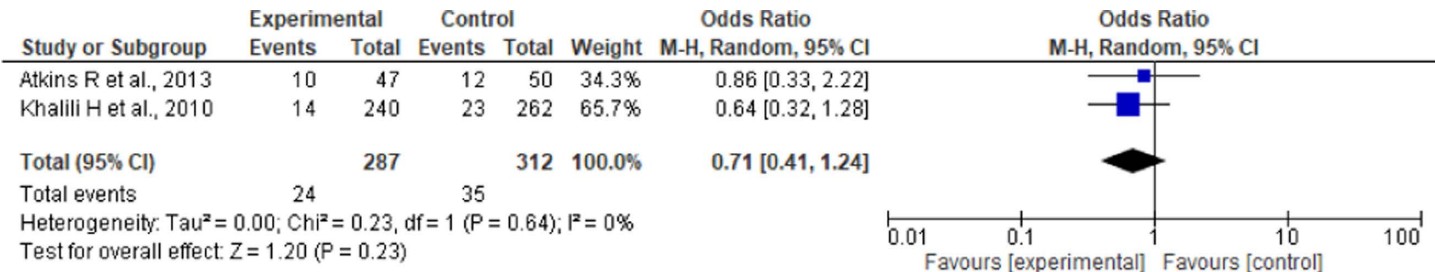

**Fig 4. The forest plot illustrates the Odds Ratio for the use of antiulcer drugs for stress ulcer prophylaxis (SUP) among the experimental and control participants.** The size of the data markers was based on the weight from the random effects analysis, with CI representing the confidence interval.

hospital costs [26,27]. Despite the wide 95% CI (0.97 to 20.80), likely due to differences in study design, patient populations, or pharmacist interventions [28] and high heterogeneity ($I^2$ = 89%), the findings advocate for standardized approaches in future research. Nonetheless, the pooled analysis provides compelling evidence for the beneficial role of pharmacists in promoting rational anti-ulcer medication use, emphasizing their integration into multidisciplinary healthcare teams to optimize patient outcomes, consistent with Bayraktar et al., 2024 [29].

Our analysis revealed that pharmacist interventions reduced both the cost and duration of anti-ulcer therapies. Studies by Xin C et al., Chen Q et al., and Buckley et al., [16,18,23] confirmed these reductions, illustrating the economic and therapeutic benefits of pharmacist involvement. Similarly, Lankford et al., (2021) found that 547 interventions led to cost avoidance of $1,508,131 [30], and another study indicated that pharmacist interventions reduced hospital stay [31]. Pooled estimates showed a significant impact with a standardized mean difference of -0.72 in therapy duration compared to controls, and a 95% CI of -1.16 to -0.29 (P = 0.001) [32]. Despite high heterogeneity ($I^2$ = 90%), likely due to differences in healthcare settings, methods, or patient populations [28], the consistent reduction in therapy duration and costs across studies supports the integration of pharmacists into healthcare teams to optimize resource use and enhance cost-effectiveness of anti-ulcer treatment protocols [31].

The third outcome of our analysis examined the improvement in the appropriate use of drugs for SUP, especially PPIs and other acid-suppressive therapies. Studies by Atkins

Table 6. Description of outcomes of pharmacist interventions reported in included studies.

| Author | Number of Interventions performed | Rational use indicators | Drug cost | Adverse events reported | Drug Interactions/ Drug switch | Duration of treatment | Physician's Level of Acceptance |
|---|---|---|---|---|---|---|---|
| Xin et al., 2018 | Out of 285 individuals who completed the study, a total of 143 interventions were carried out | The percentage of patients with rational indications was found to be 96.5%, and the accuracy rate of the administration route improved to 99.3% | The total cost reduction for PPIs amounted to $11,614.19 | Clostridium official associated diarrhea, respiratory infections, Hypomagnesia, Psychiatric symptoms | NR | The duration of therapy was shortened to 7.9 days | Out of 322 recommendations sent, 288 (89.4%) were accepted |
| Yailian AL et al., 2022 | A total of 29,694 interventions were carried out | The majority of drug-related problems were identified as drugs being used without an indication (18.3%) and inappropriate routes of administration (16.3%) | NR | NR | Drug switching was performed in 35.9% of the interventions | NR | Out of 23,688 interventions, 18,919 (79.9%) of the recommendations were accepted |
| Chen Q et al, 2022 | A total of 538 interventions were conducted | The rational use of medications increased to 68.45%, and the overall adherence to medication indications was 88.8% | The drug and PPI costs were significantly reduced in the intervention group, amounting to ¥2,517.78 | NR | Clinical pharmacist interventions had a significant effect on reducing unindicated drug use, inappropriate drug selection, and the use of unsuitable drug formulations | NR | NR |
| Musuuza JS et al., 2021 | A total of 155 interventions were carried out | Weekly PPI prescriptions were reduced by 0.5 | NR | NR | NR | NR | NR |
| Weng A et., 2022 | A total of 53 intervention were made | Awareness of peptic ulcer knowledge and medication compliance (3.85) was higher in the intervention group. The *H. pylori* eradication rate was 82.61%. | NR | NR | NR | NR | NR |
| Atkins R et al., 2013 | A total of 1089 intervention were made | Enhancements were observed in the proper use of PPIs for stress ulcer prophylaxis, a reduction in chronic PPI therapy, and increased documentation of PPI indications | NR | NR | NR | A reduction in the appropriate duration of PPIs for stress ulcer prophylaxis was observed | NR |
| Khalili H et al., 2010 | A total of 240 intervention were made | The use of acid suppression therapy (AST) was reduced by 47.1%. Among those receiving AST without an indication for stress ulcer prophylaxis (SUP), 101 out of 113 (89.4%) had no valid indication, while among those receiving AST with an indication for SUP, 12 out of 14 (85.7%) had appropriate indications. | NR | GI bleeding | Higher proportion of Omeprazole was used | NR | NR |

*(Continued)*

**Table 6.** (Continued)

| Author | Number of Interventions performed | Rational use indicators | Drug cost | Adverse events reported | Drug Interactions/ Drug switch | Duration of treatment | Physician's Level of Acceptance |
|---|---|---|---|---|---|---|---|
| Buckley et al., 2015 | A total of 371 intervention were made | The implementation of appropriate acid suppression therapy (AST) for ICU patients with some risk factors increased to 35.4%. However, 38.9% of patients continued to receive inappropriate AST after being transferred from the ICU to the general ward. | The total cost of therapy was reduced to $1,752.21 in the ICU and $1,528.28 in the general ward. Additionally, the costs for using H2RAs and PPIs decreased to $910.18 and $842.03, respectively, for ICU patients | NR | The use of the preferred H2RA agent increased to 89.1% among ICU patients | The total duration of H2RA and PPI therapy was reduced to 4.7 ± 6.1 days | NR |
| Tandun et al., 2019 | A total of 58 interventions were made | Following the intervention, 62.5% (30 out of 48) of residents from both facilities received a deprescribing order. In Facility 1, 100% (20 out of 20) of residents successfully underwent PPI deprescribing, while Facility 2 had a lower success rate of 80% (8 out of 10) | NR | Heartburn symptoms | Tapering the PPI dose before discontinuation was observed in 63.3% of residents (23 out of 30). Other approaches included discontinuing the PPI without tapering (16.7%, or 5 out of 30), maintaining a reduced PPI dose (13.3%, or 4 out of 30), and switching to ranitidine (6.7%, or 2 out of 30) | NR | NR |

NR= Not reported, AST= Acid suppression therapy, SUP= Stress Ulcer Prophylaxis, H2RA = histamine-2 receptor antagonist, ICU= Intensive care unit.

et al,. and Khalili et al,. [21,22] reported significant enhancements in rational drug use following pharmacist interventions. This underscores the critical role of pharmacists in optimizing anti-ulcer drug selection and usage for SUP based on clinical guidelines. The pooled analysis of these studies showed an odds ratio of 0.71, indicating a positive trend towards improved rational drug use with pharmacist involvement compared to the control group. However, the CI of 0.41 to 1.24 shows no statistical significance (P = 0.23), and the wide range suggests some uncertainty in the effect size. Low heterogeneity ($I^2$ = 0%) indicated consistency across studies. Despite the lack of statistical significance, the consistent improvement implies that pharmacist interventions may still promote more appropriate SUP therapy [28]. Further large-scale studies are needed to confirm these findings and clarify the role of pharmacists in enhancing the rational use of anti-ulcer medications for SUP. These results highlight the potential value of integrating pharmacists into health care teams to improve treatment decisions and reduce inappropriate acid-suppressive therapy use.

Physicians' acceptance of pharmacist interventions is essential for successful medication management. Two studies in our review showed high acceptance rates of 79.9%–89.4% for pharmacist recommendations, highlighting the collaborative potential in patient care optimization, especially in anti-ulcer therapies. Zaal et al., (2020) similarly reported a 71.2% acceptance rate by physicians for pharmacist interventions [28]. These high rates indicate that physicians value pharmacists' expertise in addressing medication-related issues, such as drug selection, dosing, and adherence to guidelines. Collaboration is crucial for enhancing patient outcomes by leveraging the complementary skills of healthcare providers. These findings emphasize the need for a multidisciplinary approach in healthcare, integrating pharmacists in medication management, particularly in complex areas, such as peptic ulcer disease [29,30]. Continued collaboration between pharmacists and physicians is vital to improve medication use and patient safety [31].

## Strengths and limitations

The study's rigorous methodology, following the PRISMA guidelines and comprehensive assessment of pharmacist-led interventions for peptic ulcer disease provide valuable insights into their impact on medication use and patient outcomes, while the inclusion of diverse interventions strengthens the findings. This review had several limitations that may affect its applicability and validity. The exclusion of non-English studies potentially introduces bias and reduces generalizability. Heterogeneity is likely increased by variations in study designs, small sample sizes, and diversity in pharmacist interventions. The predominance of observational studies, which are susceptible to selection bias and confounding factors, could also impact the overall reliability of the findings.

## Conclusion

This review highlights the significant impact of pharmacist interventions in the management of anti-ulcer therapies, particularly in improving rational drug use, reducing treatment costs and duration, and enhancing appropriate medication use for SUP. Pooled analysis of studies demonstrated that pharmacist involvement resulted in better medication adherence and corrected irrational drug use, with positive trends in therapy duration and cost reduction. Overall, these findings advocate for a more prominent role for pharmacists in peptic ulcer management as their interventions improve clinical outcomes, optimize resource use, and foster a collaborative approach in healthcare settings. Further research using standardized methodologies could strengthen the evidence for these benefits.

## Supporting information

**S1 File. Search strategy in database.**
(DOCX)

**S2 File. List of all study searched.**
(XLSX)

**S3 File. List of all excluded study with explanation.**
(XLSX)

**S4 File. List of all included study with explanation.**
(XLSX)

**S1 Checklist. PRISMA 2020 checklist.**
(DOCX)

## Acknowledgment

We would like to acknowledge the Madan Bhandari Academy of Health Sciences, Hetauda, Nepal, for supporting us in conducting this study.

## Author contributions

**Conceptualization:** Biswash Sapkota.

**Formal analysis:** Biswash Sapkota.

**Methodology:** Biswash Sapkota, Bipindra Pandey, Keshav Dhakal.

**Resources:** Bipindra Pandey.

**Software:** Bishal Sapkota, Keshav Dhakal.

**Supervision:** Bijay Aryal.

**Writing – original draft:** Biswash Sapkota.

**Writing – review & editing:** Biswash Sapkota, Bishal Sapkota.

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
