## [Decision Letter · Decision Letter 0]

15 Nov 2024

PONE-D-24-43608A Systematic Review and Meta-Analysis on Pharmacist-Led Interventions for the Management of Peptic Ulcer DiseasePLOS ONE

Dear Dr. Sapkota,

Thank you for submitting your manuscript to PLOS ONE. After careful consideration, we feel that it has merit but does not fully meet PLOS ONE’s publication criteria as it currently stands. Therefore, we invite you to submit a revised version of the manuscript that addresses the points raised during the review process. Please submit your revised manuscript by Dec 30 2024 11:59PM. If you will need more time than this to complete your revisions, please reply to this message or contact the journal office at plosone@plos.org . Please include the following items when submitting your revised manuscript:

We look forward to receiving your revised manuscript.

Kind regards,

Obed Kwabena Offe Amponsah, PharmD, Ph.D.

Academic Editor

PLOS ONE

2. Please include a separate caption for each figure in your manuscript.

3. As required by our policy on Data Availability, please ensure your manuscript or supplementary information includes the following:

Additional Editor Comments:

Thank you for presenting this relevant research work for publication. Although well-written, please pay attention to grammar and spelling errors in the manuscript. For instance, Newcastle on line 110 is misspelled.

Reviewers' comments:

Reviewer's Responses to Questions

**Comments to the Author**

1. Is the manuscript technically sound, and do the data support the conclusions?

Reviewer #1: Partly

Reviewer #2: Yes

2. Has the statistical analysis been performed appropriately and rigorously?

Reviewer #1: Yes

Reviewer #2: Yes

3. Have the authors made all data underlying the findings in their manuscript fully available?

Reviewer #1: Yes

Reviewer #2: Yes

4. Is the manuscript presented in an intelligible fashion and written in standard English?

Reviewer #1: Yes

Reviewer #2: Yes

5. Review Comments to the Author

Reviewer #1: Thank you for the opportunity to review this manuscript. I have a few comments:

1. The introduction is long and could be more concise

2. Provide additional details on managing high heterogeneity (e.g., subgroup analysis), which could clarify if the variability across studies affects specific outcomes.

3. The interpretation of the results could be made more accessible for broader readership

4. The authors should standardize terms used throughout the results (e.g., referring to “rational drug use” vs. “appropriate medication use”) for better coherence.

5. Some sentences are lengthy, particularly in the Methods and Discussion sections. Breaking these down would improve readability.

6. Minor gramatical error

Reviewer #2: #23 databases for studies published from their inception up to June 2024.Please be specific with the date instead of broad statement ‘’their inception’’.

Exclusion criteria

#88 Do the authors think that excluding articles not published in English can introduce publication bias?

What are the limitations to this study? The authors need to clearly articulate the limitations (strengths and weaknesses of this study).

Figure 1: Flow diagram of study selection (Page 5) The authors should please adjust the flow diagram to reflect 9 boxes with clear description (Page MJ, McKenzie JE, Bossuyt PM, Boutron I, Hoffmann TC, Mulrow CD, et al. The PRISMA 2020 statement: an updated guideline for reporting systematic reviews. BMJ 2021;372:n71. doi: 10.1136/bmj.n71)

6. PLOS authors have the option to publish the peer review history of their article (what does this mean? ). If published, this will include your full peer review and any attached files.

**Do you want your identity to be public for this peer review?** For information about this choice, including consent withdrawal, please see our Privacy Policy .

Reviewer #1: No

Reviewer #2: **Yes: ** Dr Emmanuel Babafemi

---

## [Author Response · Author response to Decision Letter 1]

13 Dec 2024

A separate file has been uploaded that responds to all the queries of the reviewers.

---

## [Decision Letter · Decision Letter 1]

17 Feb 2025

A Systematic Review and Meta-Analysis on Pharmacist-Led Interventions for the Management of Peptic Ulcer Disease

PONE-D-24-43608R1

Dear Dr. Biswash,

We’re pleased to inform you that your manuscript has been judged scientifically suitable for publication and will be formally accepted for publication once it meets all outstanding technical requirements.

Kind regards,

Obed Kwabena Offe Amponsah, PharmD, Ph.D.

Academic Editor

PLOS ONE

Additional Editor Comments (optional):

Reviewers' comments:

Reviewer's Responses to Questions

**Comments to the Author**

1. If the authors have adequately addressed your comments raised in a previous round of review and you feel that this manuscript is now acceptable for publication, you may indicate that here to bypass the “Comments to the Author” section, enter your conflict of interest statement in the “Confidential to Editor” section, and submit your "Accept" recommendation.

Reviewer #2: All comments have been addressed

2. Is the manuscript technically sound, and do the data support the conclusions?

Reviewer #2: Yes

3. Has the statistical analysis been performed appropriately and rigorously?

Reviewer #2: Yes

4. Have the authors made all data underlying the findings in their manuscript fully available?

Reviewer #2: Yes

5. Is the manuscript presented in an intelligible fashion and written in standard English?

Reviewer #2: Yes

6. Review Comments to the Author

Reviewer #2: I have gone through the authors response to my comments and they have actually addressed my concerns

7. PLOS authors have the option to publish the peer review history of their article (what does this mean? ). If published, this will include your full peer review and any attached files.

**Do you want your identity to be public for this peer review?** For information about this choice, including consent withdrawal, please see our Privacy Policy .

Reviewer #2: No

---

## [Editor Report · Acceptance letter]

PONE-D-24-43608R1

PLOS ONE

Dear Dr. Sapkota,

I'm pleased to inform you that your manuscript has been deemed suitable for publication in PLOS ONE. Congratulations! Your manuscript is now being handed over to our production team.

Kind regards,

on behalf of

Dr. Obed Kwabena Offe Amponsah

Academic Editor

PLOS ONE